# Mitochondrial Metabolism: A New Dimension of Personalized Oncology

**DOI:** 10.3390/cancers15164058

**Published:** 2023-08-11

**Authors:** Babak Behnam, Farzad Taghizadeh-Hesary

**Affiliations:** 1Department of Regulatory Affairs, Amarex Clinical Research, NSF International, Germantown, MD 20874, USA; 2ENT and Head and Neck Research Center and Department, The Five Senses Health Institute, School of Medicine, Iran University of Medical Sciences, Tehran 1445613131, Iran; 3Department of Radiation Oncology, Iran University of Medical Sciences, Tehran 1445613131, Iran

**Keywords:** mitochondria, personalized oncology, cancer stem cell, T cell

## Abstract

**Simple Summary:**

Cancer cells are dependent on normal cells for their survival and functionality because they can use nanoscale tubes to steal the mitochondria from immune cells. It also highlights the significance of mitochondria in the biology of cancer cells as the key organelles for cellular metabolism and energy generation. Recent research has shown that mitochondria are critical for cancer cell survival in the hostile tumor microenvironments, immune system evasion, acquisition of more aggressive characteristics, and treatment resistance. This article discusses the role of mitochondrial metabolism in cancer biology, customized cancer therapy, and how it affects cancer resistance to chemotherapy, immunotherapy, and radiation. For instance, by scavenging the produced reactive oxygen species, functioning mitochondria might enhance cancer resistance to radiation. According to this hypothesis, targeting mitochondria may improve oncological results. The tumors can respond completely to anticancer therapies or even experience malignant progression while receiving them. As a result, individualized cancer treatment is essential. Up until now, genetic analysis has been the foundation for customized cancer treatment. There is evidence that cancers with a high mitochondrial concentration are more difficult to cure. Evaluation of mitochondrial metabolism before therapy may supplement genetic data and enhance the personalization of oncological interventions.

**Abstract:**

Energy is needed by cancer cells to stay alive and communicate with their surroundings. The primary organelles for cellular metabolism and energy synthesis are mitochondria. Researchers recently proved that cancer cells can steal immune cells’ mitochondria using nanoscale tubes. This finding demonstrates the dependence of cancer cells on normal cells for their living and function. It also denotes the importance of mitochondria in cancer cells’ biology. Emerging evidence has demonstrated how mitochondria are essential for cancer cells to survive in the harsh tumor microenvironments, evade the immune system, obtain more aggressive features, and resist treatments. For instance, functional mitochondria can improve cancer resistance against radiotherapy by scavenging the released reactive oxygen species. Therefore, targeting mitochondria can potentially enhance oncological outcomes, according to this notion. The tumors’ responses to anticancer treatments vary, ranging from a complete response to even cancer progression during treatment. Therefore, personalized cancer treatment is of crucial importance. So far, personalized cancer treatment has been based on genomic analysis. Evidence shows that tumors with high mitochondrial content are more resistant to treatment. This paper illustrates how mitochondrial metabolism can participate in cancer resistance to chemotherapy, immunotherapy, and radiotherapy. Pretreatment evaluation of mitochondrial metabolism can provide additional information to genomic analysis and can help to improve personalized oncological treatments. This article outlines the importance of mitochondrial metabolism in cancer biology and personalized treatments.

## 1. Introduction

Cancer is a heterogeneous illness made up of various biological entities that require various therapies. Due to this problem, the world is moving away from one-size-fits-all cancer treatment regimens toward ones that are risk-adapted [1]. Recent researchers aim to identify the predictive factors influencing outcomes to personalized therapies and enhance quality of life while preserving efficacy. Predictive indicators for therapy response and toxicity are as important to illness as prognostic factors.

Cancer cells require normal cells for survival and function. By using nanoscale tube-like structures, cancer cells steal mitochondria from immune cells (CD8+ T cells and natural killer [NK] cells) [2]. Aside from providing energy, mitochondria also play a significant role in cancer cell survival and growth. Moreover, mitochondria are critical to the biology of cancer stem cells (CSCs), contributing to their resistance to chemo- and radiotherapy [3].

The purpose of this article is to provide a detailed understanding of mitochondrial function in cancer metabolism and how it is relevant to improving different cancer treatments, particularly radiotherapy (RT). RT is used in over 50% of cancer cases [4], and aims to deliver the maximum dose to the affected area while minimizing harm to healthy tissues. Each RT schedule is determined by several factors, including beam type, total and per fraction doses, treatment length, time between fractions, and dose rate. Personalized radiotherapy aims to optimize the RT schedule—per the specific tumor and host characteristics—to maximize treatment outcomes while minimizing the likelihood of adverse effects [5]. Currently, RT recommendations are mainly based on population averages obtained from studies. This paradigm has two problems: tumors are generally heterogeneous with different genetic and epigenetic signatures, and tumor hosts vary in racial, ethnic, and genetic features, which might affect the treatment outcomes [5]. Emerging evidence reflects the importance of patient characteristics, including age, gender, ethnicity, comorbidities, lifestyle, and intrinsic characteristics of cancer on treatment response [6,7,8]. This strategy has become a discipline in oncology called *Personalized Cancer Treatment*. To date, personalized oncology has been principally based on genomic analysis, using different testing, for example, next-generation sequencing (NGS) [9]. This paper illustrates how mitochondrial metabolism can serve as a predictive factor of treatment response. This additional information can improve the existing personalized treatment based on genomic analysis. The varied function of mitochondria in cancer metabolism is discussed in the following section, along with how essential healthy mitochondria are used for the survival and development of cancer.

## 2. The Pivotal Role of Mitochondria in Cancer Cells’ Metabolism

Cancer cells rely on functional mitochondria to survive in the harsh tumor microenvironment (TME), evade the immune system, progress to less differentiated types, and resist different treatment modalities [10], as follows: (Figure 1)

(A)Surviving in the TME via the following mechanism:
(A1)Metabolic switch to glycolysis: cancer cells are reorganized to tolerate the hypoxic, acidic, and hypoglycemic conditions of TME. Hypoxia-inducible factor-1α (HIF-1α) is one of the primary regulators of this metabolic alteration. In the harsh TME, HIF-1α overexpression leads to a metabolic switch from oxidative phosphorylation (OxPhos) into glycolysis. This alteration can maintain the cellular adenosine triphosphate (ATP)/adenosine diphosphate (ADP) level in the hypoxic TME. It has been demonstrated that HIF-1α relies on functional mitochondria for a secure continuous function [11]. In 2020, van Gisbergen et al. realized that cancer cells with severe mitochondrial dysfunction showed a decrease in CAIX expression and HIF-1α levels. The authors concluded that functional mitochondria are essential for the stabilization of HIF-1α [11].(A2)Scavenging reactive oxygen species (ROS): hypoxic condition of TME is associated with increased ROS production in cancer cells. When there is insufficient oxygen availability, the electron transport across the mitochondrial complexes is slowed down. This causes the electrons to leak out of the electron transport chain (ETC) and interact with oxygen, producing ROS. Functional mitochondria can detoxify the released ROS by preserving the cellular NADPH sources. This function is mediated by increased NADH production, representing mitochondrial function [12,13].(A3)Arresting cell cycle: cancer cells can tolerate the harsh TME by dormancy, which is the mitotic arrest at the G_0_/G_1_ cycle phase [14]. Cell cycle progression is regulated by a dedicated system consisting of cyclins and cyclin-dependent kinases (CDK). It has been demonstrated that mitochondria can mediate dormancy in colon cancer cells by HIF-dependent activation of p21 and p27 (two CDK-cyclin inhibitors) [11,15], in prostate cancer cells by activating the MAPK-p38 pathway [16,17], and in leukemic stem cells by activating the mTOR pathway [18,19].(A4)Maintaining pH homeostasis: In contrast to normal cells, cancer cells can tolerate acidic TME using a dedicated transmembrane glycoprotein called carbonic anhydrase IX (CA IX). This protein preserves intracellular pH by absorbing extracellular bicarbonate and sending out intracellular lactate [20,21]. It has been demonstrated that mitochondria are the upregulators of CA IX [11].(A5)Mediating autophagy: mitochondria can facilitate autophagy by raising the level of intracellular ROS, which leads to the inactivation of mTORC1 (an autophagy inhibitor) and the activation of NRF2 (an autophagy activator) [22,23,24,25].(A6)Angiogenesis: secretion of different angiogenic factors (e.g., VEGF, PGF, angiopoietin) in cancer cells is HIF-dependent [26]. Mitochondria conduct angiogenesis by securing HIF function [11].(A7)Mitochondrial hijacking: cancer cells can steal mitochondria from T cells (and NK cells) via nano-scale tubes. Saha et al. demonstrated that this process is GTP-dependent [2]. Functional mitochondria can secure mitochondria hijacking by providing GTP from their TCA cycle [27].(B)Immune evasion: completed via facilitating TME acidification, glucose influx, PD-1 upregulation on T cells (by mitochondrial hijacking) [28], recruiting myeloid-derived suppressor cells (MDSCs), PD-L1 overexpression on cancer cells (via STING-IFN pathway), MHC-1 downregulation, and the secretion of immunosuppressants [10]. Additionally, T cells’ mitochondrial hijacking leads to PD-1 upregulation on T-cells and depletes their energy to provide long-term cancer-fighting action [28].(C)Aggressiveness: mitochondria are crucial for cancer progression via mediating genomic instability, quiescence evasion, and epithelial-to-mesenchymal transition (EMT) [10]. An increase in cellular ROS is the most common promoter of these three processes. Genomic instability is mediated by an increase in ROS levels and damage to nuclear nucleosides and inducing minority MOMP (mitochondrial outer membrane permeabilization) [10]; quiescence evasion is conducted by an increase in cellular ROS and following the activation of the Ras pathway [29,30]; Section 3 summarizes how mitochondria are involved in EMT. ROS is a double-edged sword, destroying cancer cells at high levels and promoting cancer progression at moderate levels. Functional mitochondria help cancer cells to maintain cellular ROS at higher levels (so-called “elevated ROS balance”), facilitating cancer progression without damage to the cellular structures [31].(D)Treatment resistance: mitochondria can protect cancer cells from chemotherapy and RT by eliminating the released ROS. Additionally, they increase chemotherapy resistance by encouraging the function of efflux pumps (by providing ATP) and inducing cell cycle arrest. Additionally, mitochondrial hijacking from T cells impairs the long-term effects of anti-PD-1 treatment [10].

**Figure 1 cancers-15-04058-f001:**
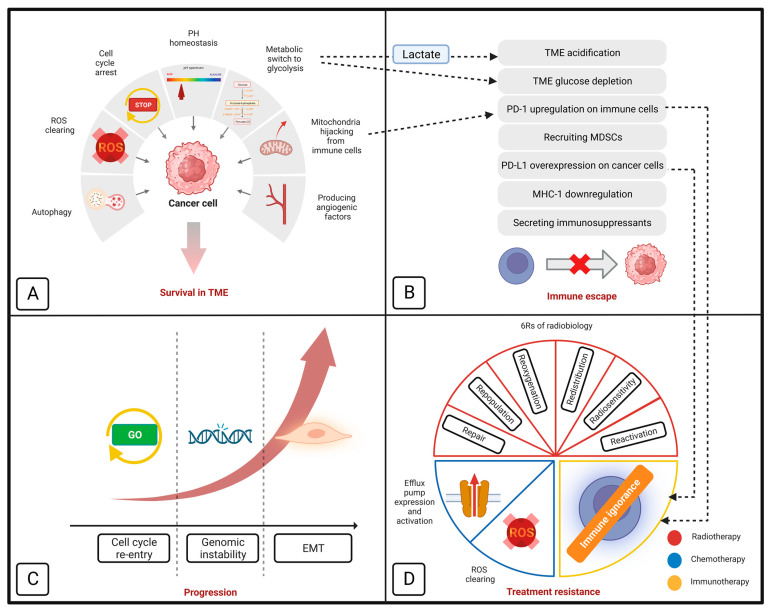
Schematic model of how mitochondria contribute to cancer cells’ survival in tumor microenvironment (**A**), immune evasion (**B**), progression (**C**), and resistance to different treatment modalities (**D**). Section D also demonstrates the importance of mitochondrial metabolism in ‘6Rs’ of radiobiology. EMT indicates epithelial-mesenchymal transition; MDSC, myeloid-derived suppressor cell; MHC-1, major histocompatibility complex class I; PD-1, programmed cell death protein-1; PD-L1, programmed cell death protein-ligand 1; ROS, reactive oxygen species; and TME, tumor microenvironment (retrieved from [10,32]).

## 3. Mitochondria Individualized Role in Cancer Metastasis

Metastasis happens in a very diverse and individualized pattern [33]. The players in the molecular pathway of metastasis and the therapeutic response to metastasis should also be considered in a personalized and idealized context. In order for cancer cells to spread, they must first undergo EMT, during which they lose intercellular adhesions and obtain high capacity for local migration, vascular invasion, and resistance to apoptotic stimuli. [34]. It has been found that there is a link between EMT and the stemness of cancer cells. These two processes are controlled by common mediators such as HIFs, SNAIL, and SLUG/SOX9 [35,36]. More functional mitochondria can promote EMT through releasing more mitochondrial ROS (mtROS), which activates different pathways, such as MAPK PI3K/Akt/mTOR, and VEGFA–SOX2–SNAI2 pathways [36,37,38]. Moreover, it is essential to acknowledge that mitochondria are directly involved in the cancer cells’ proliferation, invasion, and metastasis by enabling the linkage between β1 integrin and the extracellular matrix [39]. This process is mediated by lysyl oxidase (LOX), which requires HIF-1α for a secured function. Mitochondria can promote this process by promoting HIF-1α stability [11,40]. It is of utmost importance to employ targeted anti-mitochondrial to impede the process of EMT and curb the spread of cancer cells throughout the body. This approach can prove to be instrumental in arresting the progression of cancer and enhancing the effectiveness of treatment. Precisional targeting of cancer-specific mitochondria can reduce their ability to de-differentiate, proliferate, and metastasize, and helps to improve the treatment results and overall prognosis.

## 4. Targeting Mitochondria: A Practical Strategy for Personalized Cancer Treatment

Thanks to the developments in medical genetics and molecular biology, the function of mitochondria in several cellular functions, including apoptosis, redox balance, macromolecule production, and calcium homeostasis, has been demonstrated [41,42]. In contrast to the ancient Warburg theory, the mitochondria of cancer cells are functional, supporting their survival and function [10]. As noted earlier, mitochondria can contribute to the development, progression, and metastasis of cancer. In addition, it has a crucial role in treatment resistance. As noted in Section 2, functional mitochondria can help cancer cells to overcome chemotherapy effects by scavenging released ROS and activating multidrug resistance pumps [10]. Also, they can improve the resistance against immunotherapy, by inhibiting the immune cells’ entry to the TME by depleting the glucose content of TME, acidifying the TME, and mediating the mitochondria hijacking from immune cells [10,43]. Next, we outline how mitochondria can improve the cancer cells resistance against radiotherapy. In a recent study, Taghizadeh-Hesary et al. demonstrated that mitochondria have a contributing role in tumor response to radiotherapy. They demonstrated that mitochondria are involved in so-called 6Rs of radiobiology [32] (Figure 1). The details of this link were presented as follows:

(a) Repair: DNA damage is the primary cause of RT’s cytotoxic effects. Cancer cells with improved DNA repair mechanisms can counteract this effect. Mitochondria can support ATP-dependent proteins responsible for DNA integrity-related, including PARP-1 [44], XRCC1 [45], ATM [46], and DNA ligases [47], by providing enough ATP molecules.

(b) Repopulation: Mitochondria can support cancer cells proliferation by supplying the building materials, including nucleic acids, amino acids, and lipids through stabilizing HIF-1 and metabolic switching to glycolysis [48].

(c) Reoxygenation: HIF-1 can mediate tissue reoxygenation by promoting the expression of different angiogenic factors and shielding endothelial cells from radiation effects [49]. HIF-1 needs functional mitochondria to function properly [11]. Consequently, healthy mitochondria can aid in the reoxygenation of tumor tissue.

(d) Redistribution: Cyclin-Cdk complexes carefully control the cancer cells’ cell cycle [50]. The radiosensitive phases of the cell cycle are G2 and M and the radioresistant phases are G1 and S [51]. Cell cycle progression depends on dynamic responses of mitochondria during the G1 and S phases, when mitochondria fuse to form a hyperactive network; after that, they undergo fission to ensure proper partitioning between the two daughter cells [50]. In addition, functional mitochondria can help the cell cycle progression by supplying enough energy [52].

(e) Reactivation: Cancer cells have the ability to avoid activated immune cells by using immune inhibitory molecules like programmed death-ligand 1 (PD-L1) [53]. It has been shown that in the hypoxic TME, HIF-1α mediates PD-L1 expression on cancer cells [54]. This process is supported by functioning mitochondria which help to stabilize HIF-1α [11]. On the other hand, Akbari et al. found a direct correlation between T cell mitochondrial capacity and the expression of PD-1. If T cells have limited mitochondrial capacity, they may experience an overexpression of PD-1, ultimately leading to inactivity [28]. The study conducted by Saha et al. conclusively showed that specific nanotubes enable cancer cells to hijack the mitochondria of T and NK cells [2]. Applying this strategy, cancer cells deliberately raise their PD-L1 levels while boosting PD-1 in immune cells to cunningly evade the immune system.

(f) Radiosensitivity: Functional mitochondria can reduce the radiosensitivity of cancer cells by scavenging the released ROS and mediating the removal of damaged mitochondria, a process called mitophagy [32]. Hitherto, numerous biological factors have been linked to the intrinsic radiosensitivity of cancer cells, including p53, transforming growth factor beta (TGF-β), and isocitrate dehydrogenase 1 (IDH1) among others. For instance, p53 can improve radioresistance by enhancing the mitochondrial DNA integrity and PGC-1α (peroxisome proliferator-activated receptor γ coactivator-1α) overexpression [55,56]. For the detailed mechanisms of other corresponding factors, the readers are referred to the study by Taghizadeh-Hesary et al. [32] (Table 1).

This section illustrated how functional mitochondria can improve the tumor resistance against the various treatments. Therefore, inhibiting the cancer cells’ mitochondria can potentially improve the treatment results.

## 5. Enhancing the Normal Cells’ Mitochondria Reduces the Radiotherapy Toxicity

The way normal tissue responds to radiation is mainly influenced by its DNA repair capacity, repopulation, and radiosensitivity [101]. As previously stated, having functional mitochondria is crucial for cells to promote DNA repair and growth and to reduce RT-induced oxidative stress. Inflammation caused by radiation is an important stage in the development of normal tissue damage. It occurs when ROS is released from the damaged cells [102]. Functional mitochondria help to reduce inflammation and prevent tissue damage by effectively eliminating ROSs [103]. Hence, enhanced mitochondrial content of normal cells can effectively mitigate the adverse impact of radiation exposure. In this section, we outline how mitochondrial metabolism is connected to the known factors’ normal tissue radiosensitivity.

Recruiting genotypic and proteomic data of patients with breast or head and neck cancer, a series of proteins are recognized as a determinant for normal tissue toxicity to radiation; including CHIT1, PDGFB, STIM1, and THPO proteins as improving radiosensitivity, and SERPINC1 and SLC4A as enhancing radioresistance [104]. Mitochondrial metabolism interprets the mechanism of action of STIM1, SERPINC1, and SLC4A. STIM1 (stromal interaction molecule 1) regulates intracellular calcium level [105] and downregulates mitochondrial metabolism as its knockout leads to more metabolically active mitochondria [106]. STIM1 exacerbates radiation toxicity by preventing mitochondrial function from neutralizing the radiation-induced ROSs. Apoptosis and mitochondrial dysfunction are instead encouraged by SERPINC1 knockout because it activates the Bax pathway [107]. In the mitochondrial anti-oxidative system, SLC4 (solute carrier 4) scavenges ROS to improve radioresistance [108]. Hence, SERPINC1 and SLC4 may enhance radioresistance by enhancing mitochondrial metabolism and their capacity to scavenge ROS molecules.TGF-β overexpression increases the susceptibility of radiation-induced pulmonary fibrosis [109] and its activation affects mitochondrial respiration via impairing the mitochondrial complex IV in lung epithelial cells [110].The JAK/STAT signaling pathway in human cells is thought to provide protection against radiation. The activation of STAT3 enhances the ability of normal cells to withstand radiation by promoting the production of NADPH (which helps maintain a balanced redox state) and ATP (which helps ensure DNA stability); hence, it enhances the mitochondrial ETC in normal cells [111].Radiation toxicities are more likely to affect older people. Higher ROS production and decreased antioxidant capability in older people have been blamed for this impact [112]. As people get older, there is mounting evidence that their ability to produce ATP and NADPH is reduced because of an accumulation of mtDNA mutations and ROS damage to the mitochondrial substructures [113]. The cellular redox processes (such as glutathione) and the ATP-dependent enzymes responsible for repairing DNA damage are each impaired, necessitating NAPDH to function [114]. As a result, its relationship with radiation damage may be influenced by aging’s impact on mitochondrial metabolism.Several mechanisms have been proposed to explain how smoking during RT may increase the frequency and severity of radiation-induced acute and delayed toxicities [115]. Through endothelial damage and coagulation, it impairs tissue repair and triggers an inflammatory cascade, which increases the rate and severity of acute radiation toxicities and causes late toxicities [116]. Both acute and late radiation toxicities from tobacco smoke affect mitochondria negatively. Smoke exposure alters the mitochondrial membrane potential, which causes the release of ROS from the mitochondria and ultimately results in cellular death. DAMPs are then released into the extracellular matrix, where they connect to toll-like receptors (TLRs) on tissue macrophages and trigger the NF-kB pathway. Inflammatory cytokines are released as a result, which damages healthy tissue and exacerbates acute radiation-induced inflammation [116]. The main cause of delayed radiation toxicities, which manifest at least three months after RT, is the replacement of normal tissues by fibrotic tissues with inadequate blood flow [117]. In order for tissue regeneration and angiogenesis to be mediated by wound macrophages—the key players in wound healing—proper mitochondrial function is a crucial precondition and determining factor in the early stages of wound repair [118]. Therefore, increased radiation toxicity in smokers is justified by mitochondrial damage.

Alcohol intake can also enhance the incidence and severity of tissue fibrosis after radiation exposure, which can aggravate radiation-induced toxicities [119]. In order for macrophages to effectively repair the damaged tissue, as mentioned above, functional mitochondria are necessary [118]. Since ethanol can harm normal cells’ mitochondria by inducing oxidative stress, its detrimental effects on mitochondrial metabolism may contribute to the radiotherapy’s delayed toxicities [120]. As a result, continued use of cigarettes or alcohol during RT may each cause certain radiation-induced toxicities.

## 6. Immune Cells’ Mitochondria: A Chance to Improve Treatment Results

In addition to immunotherapy, a powerful immune system can improve the treatment results of radiotherapy and chemotherapy [32,121]. To improve the normal cells’ mitochondrial content and activity several strategies can be employed. The mitochondria quality can increase by two strategies; (1) improving the lifestyle by regular exercise [122], specific diets (low-specific dynamic action diet [123], branched-chain amino acid-rich diet [124], and Mediterranean diet [125]); good sleep [126], healthy weight [127], alcohol abstinence [128], and smoking cessation [129]; and (2) mitochondria boosting agents (e.g., coenzyme Q10, activators of adenosine AMPK, acetyl-L-Carnitine; mammalian target of rapamycin [mTOR], PGC-1α, etc.) [130,131]. In addition, the human gut microbiota is another modulator of mitochondrial fitness. It has been demonstrated that microbiota-derived metabolites are necessary for the proper action of mitochondrial metabolisms, including glycolysis, tricarboxylic acid (TCA) cycle, oxidative phosphorylation, as well as amino acid and fatty acid metabolism. The mitochondrial boosting strategies are diverse. Detailed information is presented in the following sources [131,132].

## 7. Heteroplasmy Provides Unique Profiles in Cancer

Heteroplasmy is the presence of more than one type of organellar genome (mitochondrial DNA or plastid DNA) within a cell. The amount of heteroplasmy is determined during oogenesis and is inherited from the mother. There are variations in the percentage of mutant alleles between oocytes and then between children. Heteroplasmy or the presence of at least two mtDNA variants within the single cell, and its level (the proportion of mutated mtDNA) are frequently seen with and in accompany tumor heterogeneity. One of the major challenges to understanding and elucidating the role of the variations in tumor growth is the heteroplasmy levels of the mtDNA variants. In turn, intratumor genetic heterogeneity affects personalized medicine strategies in a significant way since it can reduce the effectiveness of treatments and result in treatment resistance. It is interesting to note that numerous studies have linked heteroplasmic levels to both cancer risk and survival [133,134,135,136,137]. It would be essential to advance knowledge of the biological mechanisms at play, including proliferation, metastasis, and intratumoral heterogeneity, as well as the clinical implications of heteroplasmy, via recognizing the crucial role of heteroplasmy in cancer. The high mutation rate found in mtDNA, which is between 10 and 17 times higher than that of the nuclear genome, is explained by the lack of histones, effective DNA repair mechanisms, and closeness to reactive oxygen species (ROS) produced by the OxPhos system (mostly from Complex I and III) (nDNA) [138,139,140,141]. In humans, mtDNA is only inherited via the maternal line as a single unit called a haplotype, which may be shared by populations with similar ancestries. Factually, a set of haplotypes or a haplogroup can be used to distinguish across populations or ethnic groupings while certain haplogroups have advantages for environmental adaptation but are also linked to cancer [142,143,144,145,146,147,148,149].

The degree of heteroplasmy varies greatly between different kinds of cancer and individuals. It has been demonstrated that when tumors progressed, heteroplasmy varied amongst tissues. Based on the idea that some heteroplasmic variations are finally able to become dominant or are lost in cancer cells based on their tumor-promoting impact, a likely bottleneck process was proposed. The G1576C and G12009A mutations are the most prevalent in tumor cells compared to normal cells (7.8% versus 0.35% and 68.8% versus 0.35%, respectively) [150].

Although a very limited number of studies have been completed on the mechanisms of heteroplasmy shifting in cancer, there is proof that cell niche and the nucleus-mitochondrial environment regulate the OxPhos system’s energy performance, choosing particular mutant alleles [151]. For instance, it has been demonstrated that fumarate accumulation in renal cancer alters the mitochondrial content by inactivating core components necessary for mtDNA replication [152]. Alterations in DNA polymerase gamma (POLG) and mitochondrial transcription factor A (TFAM) expression, mutations in nDNA-encoded genes involved in mitochondrial biogenesis, nuclear and mitochondrial epigenetic modifications, as well as intrinsic and extrinsic stimuli, may all result in anomalies in mtCNVs [153,154,155]. Examples include the dysregulated expression of nuclear genes such as dynamin 1 (*DRP1*), mitofusin 1 (*MFN1*) and 2 (*MFN2*) mitochondrial fusion and fission proteins, BCL2 inter-acting protein 3 (*BNIP3*), PTEN-induced kinase 1 (*PINK1*), and hypoxia inducible factor 1 (*HIF1*), observed in lung, bladder, and breast cancers [156,157]. The role of the tumor microenvironment in altering the allelic frequencies of mtDNA mutations was also hypothesized based on an investigation of primary tumors and their distant metastasis [158]. Additionally, NOX2-derived redox signaling has been shown to be used by bone marrow stromal cells to transfer functioning mitochondria to acute myeloid leukemia blasts [159,160]. Together, these pathways may be crucial for the emergence of a tumorigenic environment-adaptive, unique response that is represented in the alteration of the allelic frequencies of mtDNA mutations. The nuclear insertions of mitochondrial origin (NumtS), which have been linked to cancer, should be considered in the next investigations on heteroplasmy. NumtS or mtDNA segments integrated into the nucleus during evolution are thought to occur at a rate of ~5 × 10e^−6^ per germ cell every generation [161].

Currently, methods based on mitochondrial gene editing have been proposed as a treatment choice for reestablishing the OxPhos system in conditions brought on by mtDNA mutations. A possible therapeutic target for cancer has been suggested to include components involved in mitochondrial biogenesis and metabolism [162,163,164]. Overall, in the context of personalized oncology, the importance of heteroplasmy lies in its potential implications for diagnosis, prognosis, and treatment of certain types of cancers. Heteroplasmy analysis can contribute to tracking heterogeneity within a tumor and monitoring the clonal evolution of mtDNA mutations; it may also guide treatment decisions and the development of personalized therapeutic strategies. Heteroplasmy analysis can be performed using non-invasive methods, such as liquid biopsies, which involve analyzing the circulating tumor DNA (ctDNA) or cell-free DNA (cfDNA) shed by tumors into the bloodstream.

## 8. Practical and Potential Methods to Target Cancer Cells’ Mitochondria

This section outlines the examined and proposed methods to target the cancer cells’ mitochondria. The available methods can be categorized based on their target, as follows: (1) inhibiting mitochondrial metabolism, including isocitrate dehydrogenase inhibitors (IDH1/2) (ivosidenib [IDH1] and enasidenib [IDH2]), lactate dehydrogenase inhibitors (galloflavin), OxPhos inhibitors (venetoclax plus azacytidine [165], gamitrinib [166]), mitochondrial ETC inhibitors (metformin, deguelin, and rotenone [complex I], and oligomycin and gboxin [complex V]) [167], (2) inhibiting mitochondrial upregulators, including mTOR inhibitors (temsirolimus) [168], (3) inhibiting mitochondrial protein translation (tigecycline) [169], and (4) mitochondrial apoptosis inducers, including BH3-mimetics [170]. In this regard, an emerging study (OPTIMUM, NCT04945148) aims to evaluate the impact of adding metformin (an OxPhos inhibitor) in improving the efficacy of available standard treatment in patients with glioblastoma (radiotherapy plus temozolomide). Table 2 outlines the list of clinical trials evaluating the impact of antimitochondrial therapy on radiotherapy efficacy.

As noted in Section 2, Saha et al. demonstrated that cancer cells can improve their mitochondrial content by hijacking mitochondria from immune cells [2]. In addition, cancer cells can enhance their neighboring cells’ malignancy by transferring mitochondria between themselves. This behavior has been demonstrated by Lu et al. in bladder cancer cells [171]. Saha et al. demonstrated that the Ras/Rho GTPase signaling is actively implicated in the nanotube formation. They showed that a farnesyltransferase and geranylgeranyltransferase type 1 inhibitor (L-778123) could effectively inhibit the nanotube formation [2]. Therefore, L-778123 (and similar agents) can effectively reduce the mitochondrial content of the cancer cells.

In addition, nanotechnology-based platforms have been applied to target cancer cells’ mitochondria. In comparison with non-mitochondria-targeting approaches, mitochondria-targeting nanomaterials have overcome the limitations of photodynamic therapy (PDT) (e.g., hypoxia) and photothermal therapy (PTT) and have improved the penetration and intra-mitochondrial accumulation of chemotherapeutics [172]. For instance, in 2020, Xu et al. introduced a polypyrrole-silica (Py@Si)-based hybrid nanoparticle to improve doxorubicin accumulation in CD44^+^ cancer cells [173]. To successfully deliver nanocarriers into the mitochondria, it is crucial to consider the challenges posed by the hydrophobic and double membrane of mitochondria, as well as its highly negative potential [174]. To overcome these barriers, several strategies have been applied in the design of nanocarriers. For example, adding lipophilic cations, peptides, or aptamers to polymeric nanoparticles can enable them to penetrate the mitochondrial matrix [175].

A novel approach to target the cancer cells’ mitochondria is put forward here. Taking a look at the mitochondrial ETC, another mitochondrial-targeting approach can be considered. Based on the electromagnetic principles, electric flux creates a magnetic field surrounding it [176] (Figure 2):B = μ_o_I/2πr(1)
where B is Magnetic field; I is Current; r is Distance from the conductor; and μ_o_ is Permeability of free space (=4π × 10^−7^ N/A2).

Hence, we may envision a magnetic field surrounding a mitochondrion due to its active ETC. Studies have shown that cancer cells’ mitochondria have a stronger electron flux in their ETC compared to normal cells, which aids in responding to their metabolism [177]. This characteristic can serve as an opportunity to target cancer cells’ mitochondria by an extrinsic therapeutic magnetic field, with specific intensity sparing the normal cells’ mitochondria. Electrons generated from different metabolic processes are channeled towards the mitochondrial ETC to support the cellular metabolic pathways. By disrupting the cancer cells’ mitochondria, their ETC become impaired (reverse direction of Formula 1). This effect can eventually turn off the power button of cancer cells’ mitochondria and impede their support on cancer cells’ metabolism and TME. This concept was applied in a recent in vitro study by Sharpe et al. The investigators demonstrated that applying oscillating magnetic fields with appropriate field strength, frequency, and on/off profiles could effectively arrest the cancer cells ETC, even in a nondividing status [178]. Considering the following information, the impact of external magnetic fields can be more potent in in vivo studies. Several studies have demonstrated that low-frequency external magnetic fields can modulate the tumor immune microenvironment and improve the antitumor immune response. Nie et al. demonstrated that the low-frequency magnetic field could enhance the survival of melanoma and hepatocellular murine models by reducing the number of regulatory T cells and increasing the number of CD8+ T cells and dendritic cells in TME [179,180]. It has been demonstrated that alternating magnetic fields can enhance ROS release from immune cells [181], representing mitochondrial metabolism [182]. These benefits were not reported for static magnetic fields [181]. As mentioned earlier, cancer cells’ functional mitochondria can make it more difficult for immune cells to penetrate the TME due to its harsh conditions, such as acidity, low glucose, and hypoxia. Therefore, cancer’s mitochondria-targeting magnetic field can remove the cancer cells’ support on TME modification and can increase the chance of cancer treatment. When cancer cells are exposed, their metabolism is disrupted and their support on the TME is reduced. Therefore, alternating magnetic fields can both weaken the tumor cells and activate the immune cells and facilitate their infiltration into TME to defeat the weakened cancer cells. These beneficial effects can potentially improve the response to different treatment modalities, given the importance of immune reactivation in radiotherapy [183], chemotherapy [121], targeted therapies [184], and immunotherapy [28]. This concept is in its preliminary phases and enfaces several shortcomings. For example, it is still unclear to what degree mitochondria contribute to the potential therapeutic effect of rotating external magnetic fields in cancer cells. In order to fully comprehend how magnetic fields, affect cancer cells, it is necessary to conduct multidisciplinary research that combines experimental studies and theoretical modeling (Figure 2).

## 9. Conclusions

This article illustrated how mitochondria is involved in the tumor response to different treatments as well as the normal tissue toxicity. Ever since, personalized treatment has been primarily based on genomic analyses. This paper put forward that considering the mitochondrial metabolism status of the cancer cells can provide additional information in selecting the appropriate treatment. With this concept in mind, future works can design more personalized treatments to improve the treatment results with fewer toxicities. Heteroplasmy analysis in personalized oncology provides insights into the genetic landscape of tumors, helps predict clinical outcomes, guides treatment decisions, and offers opportunities for the development of personalized therapeutic approaches. However, further research is needed to fully understand the impact of heteroplasmy and optimize its clinical utility in oncology.

## Figures and Tables

**Figure 2 cancers-15-04058-f002:**
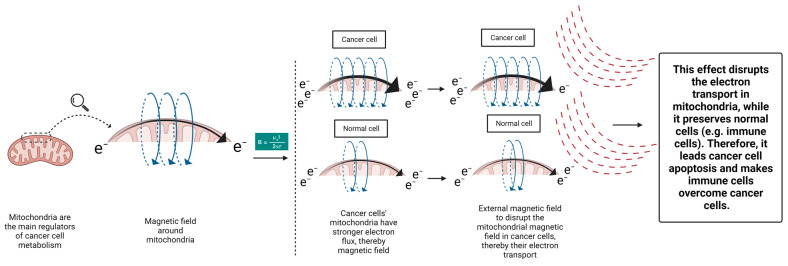
The proposed method to target cancer cells’ mitochondria using magnetic fields.

**Table 1 cancers-15-04058-t001:** The biological factors of radioresistance from the mitochondria perspective.

Factors	Cancer	Ref.	Interaction with Mitochondria	Ref.
Increasing radioresistance				
Mutated P53	Various	[57]	− Mutated p53 preserves mtDNA integrity − Mutated p53 improves mt capacity (PGC1α-mediated) − More functional mt scavenge more RT-induced ROS	[55] [56] [10]
TGF-β	HCC	[58]	− TGF-β signaling in CAFs mediates reverse Warburg effect − CAFs’ lactate and pyruvate feed cancer cells’ mt OxPhos − Activated OxPhos helps to restore NADPH − NADPH supports the antioxidant defense system	[59] [60] [61] [62]
IDH1	Glioblastoma	[63]	− Mutated IDH1 enhances mt OxPhos (ROS generation) − Mutated IDH1 downregulates cytochrome c − Cytochrome c can nullify ROS − Thus, IDH1 mutation disrupts the ROS balance	[64] [65] [66]
PARP	Breast Ovarian Prostate Pancreatic HCC	[67] [68]	− PARP requires RAD51 for HR − BRCA2 regulates RAD51 function − BRCA2 requires mt support − Thus, functional mt improves radioresistance by mediating HR	[69] [69] [70]
PI3K/Akt/mTOR pathway	Prostate	[71]	− mTOR upregulates mt proteins responsible for mt metabolism − More functional mt scavenge more RT-induced ROS	[72] [10]
Wnt/β-catenin pathway	Esophageal SCC	[73]	− Wnt upregulates HMGB1 − HMGB1 activates mitochondria − More functional mt scavenge more RT-induced ROS	[73] [74] [10]
NF-κB pathway	Breast Glioma HCC Melanoma NSCLC	[75]	− Enhances mt respiration − Regulates mt dynamics − Regulates mt gene expression	[76]
8-oxo-dG	Esophageal Gastric	[77]	− Serum 8-oxo-DG level represents cellular ROS − Cellular ROS is dependent on mt metabolism	[77] [10]
ATM	Glioma	[78]	− Preserves mtDNA	[79]
XRCC1	NSCLC HNC	[80]	− Preserves mt respiratory chain	[81]
NOTCH2	NSCLC	[82]	− Regulates mitochondrial function	[83]
KEAP1	NSCLC	[82]	− Regulates mitochondrial function − Regulates mitophagy	[84] [85]
FGFR1/3	NSCLC	[82]	− Regulates mitochondrial energy metabolism	[86]
HOTAIR	Breast	[87]	− Regulates mitochondrial function	[88] [89]
AMPK	Glioblastoma	[90]	− Preserves mt biogenesis upon energy stress	[91]
RPA1	Glioblastoma	[92]	− Preserves mtDNA	[93]
RSK2	NSCLC	[94]	− Stimulates mt OxPhos	[95]
LAPTM4B	NPC	[96]	− Activates mTOR − mTOR upregulates mt proteins responsible for mt metabolism − More functional mt scavenge more RT-induced ROS	[97] [72] [10]
Decreasing radioresistance				
TNFα	NSCLC	[98]	− Impairs mt complex I and III − Complex III is essential for NADPH activity − Thus, reduces mt capacity to scavenge RT-induced ROS	[99] [100]

Note: This Table is retrieved from the Taghizadeh-Hesary et al. study [32]. Abbreviations: 8-oxo-dG, 8-hydroxy-2′-deoxyguanosine; Akt, protein kinase B; AMPK, serine/threonine kinase AMP-activated protein kinase; ATM, ataxia-telangiectasia mutated; BRCA2, breast cancer gene 2; CAF, cancer-associated fibroblasts; FGFR1/3, fibroblast growth factor 1/3; HCC, hepatocellular carcinoma; HMGB1, high mobility group box 1; HOTAIR, HOX transcript antisense RNA; HR, homologous recombination; IDH1, Isocitrate dehydrogenase 1; KEAP1, Kelch-like ECH-associated protein; LAPTM4B, lysosome-associated transmembrane protein 4B; mt, mitochondrial; mTOR, mammalian target of rapamycin; NADPH, nicotinamide adenine dinucleotide phosphate; NF-κB, nuclear factor κB; NOTCH2, neurogenic locus notch homolog protein 2; NPC, nasopharyngeal carcinoma; NSCLC, non-small cell lung cancer; OxPhos, oxidative phosphorylation; PARP, poly (ADP-ribose) polymerase; PGC-1α, peroxisome proliferator-activated receptor-gamma coactivator 1α; PI3K, phosphoinositide 3-kinases; ROS, reactive oxygen species; RPA1, replication protein A1; RSK2, ribosomal S6 kinase; RT, radiotherapy; SCC, squamous cell carcinoma; TGF-β, transforming growth factor β; TNFα, tumor necrosis factor α; XRCC1, X-ray repair cross complementing 1.

**Table 2 cancers-15-04058-t002:** Trials on the combination of radiotherapy and therapeutic targeting of mitochondrial metabolism (source http://clinicaltrials.gov).

Clinical Trial ID	Phase	Cancer	Drug Name	Target	Mechanism	Status
NCT04945148	Phase II	Malignant glioma	Metformin	Complex I	Inhibiting OxPhos	Recruiting
NCT04275713	Phase II	Cervical cancer	Metformin	Complex I	Inhibiting OxPhos	Recruiting
NCT04732065	Phase I	Diffuse midline glioma Glioblastoma Recurrent ependymoma	ONC206	TRAIL-induced activation of ClpP	Inhibiting OxPhos	Recruiting
NCT05136846	Phase I	Non-small cell lung cancer	Papaverine	Complex I	Inhibiting OxPhos	Recruiting
NCT05325281	Phase I	Pancreatic adenocarcinoma	Devimistat	α-KGDH and PDH	Inhibiting Krebs cycle	Recruiting

Abbreviation: α-KGDH, α-ketoglutarate dehydrogenase; ClpP, mitochondrial caseinolytic protease P; OxPhos, oxidative phosphorylation; PDH, pyruvate dehydrogenase; TRAIL, TNF-related apoptosis-inducing ligand.

## Data Availability

Not applicable.

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
