# Peer review of "Mitochondrial Metabolism: A New Dimension of Personalized Oncology"

_cancers, 2023, doi:10.3390/cancers15164058_

Round 1
Reviewer 1 Report
It is an interesting review in which the authors intended to show the involvement of mitochondria in cancer response to treatments. However, I think that their involvement in the development and maintenance of tumors should be better and deeply described.
Moreover, references should be added. For example, on page 2, subtitle 2, item C, not even 1 reference was shown. The same problems can be found elsewhere in the text.
Reviewer 2 Report
This review paper focused on the role of mitochondria on cancer resistance to therapies. This topic is of wide interest and deserves a constructive discussion.
1. Figure 1 has already been reported in prior publication (7). Authors should provide an original figure.
2. Discuss details in the role of mitochondria on HIF1 function.
3. Authors should more describe how to control mitochondrial function to improve cancer treatment by anti-mitochondrial therapy.
4. Correct a mistake in line 247 “employed y”.
Round 2
Reviewer 1 Report
The authors have appropriately altered and improved the text.
